# Stop and think: Additional time supports monitoring processes in young children

**Sophie Wacker** *, **Claudia M. Roebers**

Department of Psychology, University of Bern, Bern, Switzerland

* sophie.wacker@unibe.ch

**Data Availability Statement:** All relevant data are within the manuscript and its Supporting Information files.

**Funding:** The author(s) received no specific funding for this work.

## Abstract

When children evaluate their certainty, monitoring is often inaccurate. Even though young children struggle to estimate their confidence, existing research shows that monitoring skills are developing earlier than expected. Using a paired associates learning task with integrated monitoring, we implemented a time window to—"Stop and Think"—before children generated their answers and evaluated their confidence in the chosen response. Results show that kindergarten and second grade children in the—"Stop and Think"—condition have higher monitoring accuracy than the control group. Implementing a time window thus seems to support children in their evaluation of different certainty levels. Relating individual differences in independently measured inhibitory control skills revealed a correlation between monitoring and inhibition for kindergarteners.

## Introduction

Metacognition research consistently reveals that young children show inaccurate monitoring skills. That is, they are overly optimistic when evaluating their performance. Accurate monitoring is important for a wide range of cognitive domains, including academic achievement [1]. Although often overoptimistic, young children have been found to be able to accurately monitor their performance (for example, in everyday life asking back when ambiguous information is provided or in play situations hesitating when executing an ambiguous demand [2–4]. The current approach tests the possibility that children's inaccurate monitoring is—at least in part—due to young children not taking enough time to engage in monitoring processes actively. We will explore this question in two ways. For one, we will experimentally induce a time window during which children are asked to monitor and compare different responses regarding their likelihood of being the correct answer. For another, we will independently quantify participants' inhibitory control skills and relate them to their monitoring ability.

Monitoring is a fundamental part of metacognition [5, 6], describing an individual's capability to reflect and supervise cognitive processes [4]. There are several methods to measure different monitoring aspects. Monitoring processes can be measured before a memory test and are called prospective judgments. These include, for example, judgments of learning or feelings of knowing [7]. The present study focused on retrospective monitoring processes, measured after a memory test, and described as confidence judgments. Children, adolescents, and adults experience and report their confidence on different levels, ranging from very

**Competing interests:** The authors have declared that no competing interests exist.

unsure to very sure. Thereby, kindergarten and primary school children have more difficulties estimating their performance accurately than older individuals [3, 8–10]. That is, they report being really sure, often independent of their response's accuracy. This overconfidence is partly due to imprecise monitoring skills [11–14]. Even for incorrect answers, young children often give high confidence judgments, suggesting their ability to reflect on certainty is far from fully developed at this age. During the primary school years, monitoring becomes more sophisticated and differentiated and shows an increasing congruency with actual performance [15, 16].

The theoretical background of the present approach is a broader conceptualization of higher-order cognitive self-regulation, entailing metacognition and executive functions [17, 18]. Self-regulatory skills are increasingly recognized to embrace executive functions (EF) and monitoring [19–23]. Executive functions are top-down regulating processes and include updating, shifting, and inhibition aspects [24]. Especially inhibitory control skills are needed for many everyday tasks, including learning and monitoring [25]. In younger children, inhibitory control and metacognition are assumed to be connected interactively. For example, recent results reported by Kälin and Roebers [21] uncovered an association between monitoring and executive functions. The authors showed that better inhibition was related to longer latencies when giving confidence judgments in a paired associates learning and recognition paradigm. This suggests that children with better inhibitory skills took more time to report their confidence in the selected answer. Most interestingly, these longer latencies resulted in more accurate monitoring judgments. In other words, children's monitoring accuracy was better when they took longer to generate a monitoring judgment—according to the concept of "more time to think".

With increasing age and experience, monitoring and EF are thought to differentiate and follow distinct developmental trajectories [18]. Based on their findings, Roebers et al. [22] postulated that well-developed EF are necessary to develop metacognitive skills. Deficits in monitoring processes could result from immature executive functions because a certain level of EF skills is needed to perform metacognition successfully. Therefore, the association might be stronger in younger than older children. This assumption is corroborated by findings showing that in 5- compared to 7-years-olds, monitoring is more closely related to inhibitory control skills [19]. Most recent longitudinal research addressing the interrelation of those constructs revealed that EF at an early age predicts self-regulated learning one year later but not vice versa [26]. The present study included kindergarten children and second graders to confirm previous findings: we aimed to further explore that inhibition may indeed be more critical for younger compared to older children's monitoring accuracy, with the assumption that school attendance and academic tasks gradually train children's monitoring skills [4].

From a neuropsychological perspective, inhibition might serve as a monitoring prerequisite. To engage in monitoring, the responsible neural networks need time to loop signals from the anterior cingulate cortex (ACC) to frontal structures [27]. Prefrontal structures, especially the ACC, are considered a neurological correlate for monitoring and cognitive processing [28]. Therefore, neurological monitoring signals need time for transmission [29]. Consequently, immature inhibitory skills may not provide enough time for these signals to be strengthened and passed on to related neural structures [30]. Only if an individual takes enough time to process information, monitoring aspects can come into effect [17, 31, 32]. In other words, if individuals can inhibit their prepotent responses, hesitate and ask themselves: Am I really sure about my answer?, this should benefit their monitoring accuracy [33].

However, for these processes, one must develop and experience a feeling of uncertainty. The engagement with uncertainty (carefully evaluating the own levels of certainty) may trigger metacognitive processing and can result in better performance (due to a more differentiated

and conscious evaluation) [17]. Indirect evidence supporting the view that time may play a crucial role for monitoring accuracy stems from research on the delayed JOL-effect. Delayed compared to immediate judgments-of-learning are typically more accurate, both in adults and in children [34, 35]. In cognitive tasks, for example, in a memory test, experimentally inducing a delayed response by providing additional time before responding has repeatedly been found to be an efficient means to increase the accuracy of children's responses [25, 36–38]. Simpson et al. [39] showed that if a child must wait a set time to generate an answer, this answer was more likely to be correct than answering immediately. Poor task performance can result from a prepotent response. With additional time, reflective processing may result in better performance.

None of these studies have yet applied this concept to monitoring. We will build on these findings and explore the extent to which a "Stop and Think" instruction may positively affect children's monitoring accuracy. Additionally, research focusing on the accuracy of a confidence judgment based on the prior answer showed that information processing does not end after the decision is made [40]. On the contrary, the accumulation of further information processing evolves during the interval between an answer and the corresponding confidence judgment. This accumulation may also profit from more time which is in line with our assumption. Inhibiting the prepotent response and allowing neurological signals to strengthen [41, 42] may also allow the accumulation of additional information, which may be guided by monitoring processes.

To our knowledge, no study tried to explore the influence of increased time to monitoring on children's monitoring accuracy. In an experimental setting, we implemented a delay during which the child should "Stop and Think". In the present study children solved a paired associate learning task. After studying several item pairs, subjects had to choose one out of four answer alternatives that matched the corresponding stimulus picture (recognition phase). The "Stop and Think" delay was inserted after the recognition and before the subsequent monitoring. Afterwards, children had to select a confidence judgment by rating how sure they were that they chose the correct item pair. We hypothesized that being "forced" to take more time to monitor and prevent fast and thus undifferentiated monitoring judgments would positively affect children's monitoring skills temporarily. More time until monitoring judgments are given may allow the individual to pause and reflect on the ongoing cognitive and metacognitive processes, ideally leading to better monitoring. We expected small benefits from additional time against the background of the above-mentioned findings [19, 43]. To evaluate the impact of additional time on different aspects of monitoring accuracy, we analyzed a relative (monitoring discrimination, i.e., the difference in confidence between correct and incorrect responses [44]) and an absolute score (i.e., overconfidence, the deviation of certainty from performance). We did not expect any effect on recognition as the delay was only inserted after participants had chosen an alternative.

From an individual differences perspective and in parallel to the theoretical background outlined above on the relation between inhibition and monitoring, inhibition might be a candidate factor contributing to high confidence in children, irrespective of performance [19]. One might expect that better inhibition allows the child to hesitate instead of jumping on an answer and reporting high confidence, and to reflect on the likelihood of different alternatives to be correct and thus to monitor more accurately. However, more research is needed to understand the relation between monitoring and inhibition. Despite intensive research on metacognition and its development, relatively little attention has been paid to individual differences within homogenous age groups.

The preschool and kindergarten age represents a critical time window for executive function development [45, 46]. In cognitive tasks requiring inhibitory control, findings show that

younger compared to older children benefit more from a delay [36, 47]. These results indicate that younger children need more support in inhibiting their impulsive behavior. Because of developmental maturation and still relatively immature inhibition functions [19, 48], we hypothesize that kindergarten children would benefit more from a "Stop and Think!" instruction compared to second graders.

To address the role of individual differences for monitoring beyond our experimental manipulation, we also assessed inhibitory control skills independently from metacognition. This allows us to explore the relationship between inhibition and monitoring accuracy in the control condition in which no "Stop and Think!" instruction and no delayed monitoring judgments were induced. We expected that individual differences in inhibition would be weakly but positively related to monitoring accuracy in younger but not necessarily in older (school) children in the control condition. In addition, we examined the relationship between inhibition and monitoring accuracy in the experimental group (only children with the "Stop and Think" instruction). Thus, we investigated whether there are individual differences regarding the extent to which a delay can contribute to improving monitoring accuracy. For example, children with poor inhibition might benefit more from a delay than children with already sophisticated inhibitory control skills.

## Methods

### Participants

Data stems from N = 393 children from rural and urban areas in a mid-European country. For the analysis, we recruited a sample of N = 202 (44.6% female) kindergartners between 4–6 years of age (M = 73.6 months, SD = 7.4 months) and N = 191 second graders (45.5% female) between 7–9 years of age (M = 94.2 months, SD = 7.1 months). Participants represent a sample of middle-class families mostly of Caucasian descent. The Ethics Committee of the Faculty of Human Sciences at the University of Bern approved ethical consent for the study (Approval No: 2002–100005). The parents or legal guardians of all participating children signed an informed consent. Further, all children were asked verbally to participate prior to the testing. They were further explained that they could terminate the task at any time. No child ever did. Data is entirely anonymous. Due to technical problems, we excluded N = 2 participants. Additionally, during one test session, N = 20 children had to quarantine due to COVID-19. These children were also excluded from the analyses reported below because they did not solve the paired associate learning task. Due to the current restrictions, there was no opportunity to retest them. We had to exclude N = 4 participants with an accuracy of 0% or 100% in the recognition block for the paired association task as they did not generate complete monitoring data. Restrictions due to COVID -19 and because several children had to be in quarantine, we could not examine inhibition data of all children. Therefore, the Heart and Flower task analysis is limited to an N = 330.

### Procedure and measures

Children performed two different computer-based tasks, running on tablets (Samsung Galaxy S6). During the study, trained investigators were present. Test sessions took place in a group setting in children's schools, with each participant listening to the pre-recorded instructions through headphones. Children solved a paired associates learning task with integrated monitoring (30 to 40 min.). In this task, children were to log in their answers by touching predefined areas on the screen with their index fingers. The children solved a paired association learning task encased in a cover story of two children to assess the monitoring aspect. Following a familiarization phase, the task was composed of 3 phases. In the first, the learning phase,

participants learned different numbers of item pairs (kindergarteners: 16 items, second graders: 22 items). Each item pair was presented for 4s. After the learning phase, participants solved a filler task for about 1 minute, followed by a recognition phase. Participants were shown one constituent from an item pair and had to choose one out of four possible answers as being the matching item. There was no time limit for choosing the matching item. After choosing an answer in the recognition phase, participants were immediately asked to provide a confidence judgment (CJ) for their selection in the final monitoring phase. Participants had to indicate their certainty on a 7-point Likert scale (adapted from [27]).

Children were randomly assigned to either the control group (CG; they solved the task as described above) or the experimental group (EG). Participants in the EG had to wait a set time before choosing a CJ. Research suggests that the diffusion of neurological signals to the prefrontal cortex needs about 200–250 ms [20, 30]. Other studies found that implementing a delay of 4 seconds leads to a performance improvement [36]. Therefore, we chose an interval representing a reasonable pause allowing enough time for diffusion and time for reflection. For this purpose, we implemented a fixed delay of 8s before participants could choose a CJ. Throughout this 8s delay, an animation was implemented, during which the pictures became gradually transparent and smaller. At the same time, one out of the two protagonists appeared with a big speech bubble containing the thermometer. This sequence represented the protagonists showing that they are taking time to think about their answers and their certainty following a pattern:—"Stop and Think"—All other procedures did not differ from the control group.

We generated 12 item pairs with medium difficulty (index with .57) for the kindergarteners and 15 items (index with .60) for the second graders. Item pairs for kindergarteners with a very high (index below .32, N = 2) and very low (index above .77, N = 2) difficulty and for second graders correspondingly (index below .32, N = 5; index above .77, N = 2), served as anchor items and were not used for the analysis. To address relative aspects of monitoring, we calculated a discrimination score to quantify the ability to discriminate between CJ for correct and CJ for incorrect answers [49, 50]. Additionally, we used the bias index for absolute aspects of monitoring [50]. The bias index maps to a continuous range between underestimation (-1), accurate estimation (0), and overestimation (+1).

In another session (15 min.), with a minimum delay of one week, each child solved the Hearts and Flowers task capturing inhibition and cognitive flexibility [51, 52]. For this task, two external response buttons were connected to the computer and placed on the right and left sides of the screen. In the congruent condition (heart block; N = 24 trials), a heart appeared on the right or the left side of the screen. Children had to press the button on the same side where the heart appeared. In the subsequent incongruent condition (flower block; N = 36 trials), children were to press the button on the opposite side of where the flower appeared. In the final mixed block, congruent (heart) and incongruent (flower) trials were combined and appeared in pseudorandomized order (N = 60 trials). The presentation of the stimuli was during 2500 ms, followed by an interstimulus interval of 500 ms. *Dependent Variables*. We calculated the Rate Correct Score (RCS) for every block [53], reflecting the amount of correctly solved items per second. For the Hearts and Flowers task, we excluded (N = 34) participants because overall accuracy was lower than .50 (below change level). Reaction times under 200 ms were excluded as they typically represent reflexes or second corrective responses to the previous trial. Our primary interest lay in the RCS of the flower block, which is considered to represent mainly inhibition [51].

## Statistical analysis

Our study follows a 2 (control vs. experimental group) x 2 (kindergarteners vs. second graders) between-subject design. We used Scipy, Numpy, Pandas, and StatsModel, running on Python

for data analysis, and Seaborn and Matplotlib for data visualization. Our dependent variables were examined by between-subject analysis of variances (ANOVAs) concerning monitoring. With partial eta squared ($\eta_p^2$), we estimated the effect sizes. To explore the relationship between inhibition and monitoring accuracy, we evaluated correlations analysis and reported their corresponding coefficients ($r$).

## Results

### Preliminary analysis

We conducted a between-subject ANOVA to rule out that an improvement in monitoring accuracy may be driven by primary differences in performance accuracy between the CG and EG. A significant main effect of age ($F(1,392) = 5.08$, $p = .025$, $\eta_p^2 = .013$) revealed higher performance accuracy scores for second graders ($M = .57$, $SD = .18$) -corresponding to 13 out of 22 correctly solved items—than for kindergartners ($M = .53$, $SD = .19$)—corresponding to correctly solving 9 out of 16 items. Thus, there was a well-balanced database including an about equal number of correct and incorrect answers and their confidence judgment for the monitoring analyses reported below. The main effect of the condition ($F(1,392) = 3.08$, $p = .08$, $\eta_p^2 = .008$) and the interaction ($F(1,392) = .121$, $p = .728$, $\eta_p^2 = .00$) did not reach significance. Therefore, we can assume that an improvement in monitoring accuracy observed in the EG is not an artifact of better performance accuracy.

As a preliminary analysis to evaluate the performance in the inhibition measure, we calculated a between-subject ANOVA. The main effect of age ($F(1,325) = 91.03$, $p < .001$, $\eta_p^2 = .219$) was significant, with higher correctly solved items per second for second graders ($M = .45$, $SD = .12$) than kindergarteners ($M = .33$, $SD = .11$). The main effect of the condition ($F(1,325) = 3.34$, $p = .068$, $\eta_p^2 = .01$) and the interaction ($F(1,325) = .68$, $p = .41$, $\eta_p^2 = .002$) did not reach significance. These results indicate that performance in inhibitory control skills was comparable across the CG and EG.

### Monitoring

To address relative monitoring accuracy, we evaluated the discrimination score. This score taps children's ability to metacognitively discriminate in their confidence judgments between correctly and incorrectly recognized item pairs by giving substantially higher CJ for correct than for incorrect recognition. Results of the between-subject ANOVA revealed a significant main effect of age ($F(1,389) = 14.43$, $p < .001$, $\eta_p^2 = .036$), with higher discrimination scores for second graders ($M = 1.50$ $SD = 2.53$) compared to kindergarteners ($M = .48$, $SD = 2.65$). In addition, a significant main effect of condition was identified ($F(1,389) = 4.23$, $p = .04$, $\eta_p^2 = .011$), due to participants in the EG ($M = 1.2$, $SD = 2.6$) achieving better discrimination between correct and incorrect items compared to CG ($M = .6$, $SD = 2.6$), that is, achieving more accurate monitoring (see Fig 1). The interaction did not reach significance ($F(1,389) = .04$, $p = .842$, $\eta_p^2 = .00$), thus the effect of the delay was similar in the two age groups.

As the literature offers ample evidence for young children's performance overestimation, we were also interested in an absolute score of monitoring, the bias index. This score can range from underestimation (negative values), perfect estimation (values around zero) to overestimation (positive values). The ANOVA with age and experimental condition as between-subject factor revealed a significant main effect of age ($F(1,389) = 13.71$, $p < .001$, $\eta_p^2 = .034$). Kindergartners ($M = .29$, $SD = .26$) show a stronger overconfidence compared to second graders ($M = .19$, $SD = .255$). A main effect of condition ($F(1,389) = 5.39$, $p = .021$, $\eta_p^2 = .014$) was also found, with participants in the EG ($M = .21$, $SD = .26$) showing less overconfidence compared to the CG ($M = .28$, $SD = .26$) (see Fig 2). Contrary to our hypothesis, this effect was

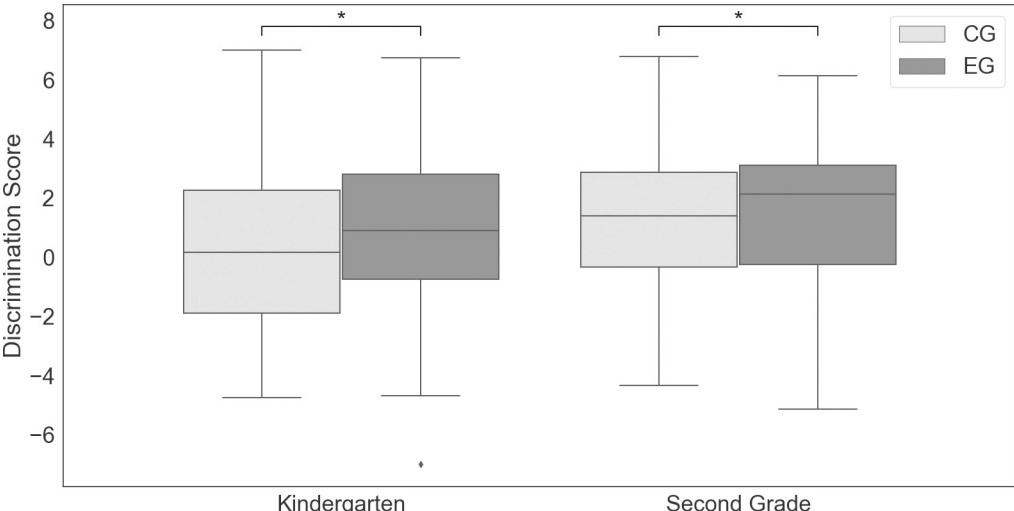

**Fig 1. Distribution of the discrimination score separated for condition and age.** Note. Boxplot for the dependent variable discrimination score, separated for Age (Kindergartners vs. Second Graders), and Condition (Control Group (CG) vs. Experimental Group (EG)). Whiskers represent 1.5 * interquartile range.

about equally strong in both age groups as the interaction did not reach significance ($F(1,389) = .382$ $p = .537$, $\eta_p^2 = .001$).

## Individual differences in inhibition

We will report the correlations separately for the two conditions. In the control group, we will explore whether individual differences in inhibition are related to monitoring accuracy (discrimination and bias index) independently from our—"Stop and Think"–manipulation. The reported results are therefore only based on participants in the CG. For this analysis, we related individual differences of the Rate Correct Score within the flower block of the Hearts and

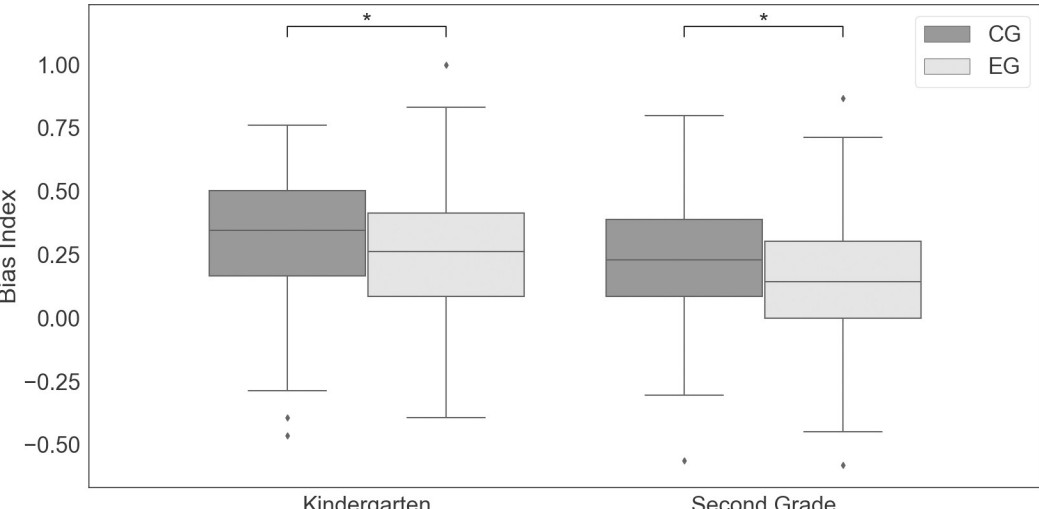

**Fig 2. Distribution of the bias index separated for condition and age.** Note. Boxplot for the dependent variable bias index, separated for Age (Kindergartners vs. Second Graders), and Condition (Control Group (CG) vs. Experimental Group (EG)). Whiskers represent 1.5 * interquartile range.

Flowers task to monitoring skills (discriminations score and bias index). For kindergarteners, correlational analysis revealed a significant positive correlation between the discrimination score and the inhibition RCS ($r$ = .225, $p$ = .034, $n$ = 88). Higher values in the discrimination score (representing more accurate monitoring) were related to better performance in the inhibition RCS (more correctly solved items per second within the flower block). This correlation only represents a small effect. Regarding the bias index and the inhibition RCS, no significant correlation was observed ($r$ = -.144, $p$ = .178, $n$ = 88). Regarding second graders, no significant correlations were observed, neither for the discrimination score ($r$ = .20, $p$ = .104, $n$ = 67) nor for the bias index ($r$ = -.061, $p$ = .626, $n$ = 67).

In the experimental condition, by correlating inhibition with our monitoring measures, we will address whether inserting the delay between recognition and monitoring has a differential effect on participants depending on their inhibitory control skills. Correlational analysis addressing the discrimination score revealed no significant correlation for kindergarteners ($r$ = .169, $p$ = .108, $n$ = 91) and second graders ($r$ = .133, $p$ = .233, $n$ = 82). Additionally, no significant correlation was found for the bias index for the kindergarteners ($r$ = -.119, $p$ = .261, $n$ = 91) as well as for second graders ($r$ = -.075, $p$ = .502, $n$ = 82).

## Discussion

The present study sheds light on young children's difficulties to accurately monitor memory performance by realizing an experimental approach and addressing individual differences in monitoring to inhibition. For one, we induced a delay between recognition and reporting confidence, and for another, we related performance in inhibition (measured with the Heart and Flower task) to our monitoring measures.

### Monitoring

In line with previous research, our results confirmed that second graders and kindergarteners already show indications of emerging monitoring skills [10, 11, 54]. In absolute terms, children were able to discriminate substantially between correct and incorrect responses, but their evaluation of incorrect item pairs was still highly overoptimistic [12]. This pattern of results underlines the still undifferentiated monitoring skills in young children [10, 55, 56].

For the relative (discrimination score) and the absolute (bias index) measure of monitoring accuracy, findings pointed into the same direction. Second graders showed a more sophisticated discrimination between CJ for correct and CJ for incorrect items and less overconfidence than kindergartners. Thus, of the age differences reported above concerning discrimination and overconfidence fit nicely into the existing literature [56–59].

As to our experimental manipulation, our results suggested that waiting and reflecting on certainty and uncertainty for the selected answer (participants in the "Stop and Think" condition) did indeed lead to better monitoring discrimination. Moreover, children who were forced to wait and reflect also showed less overconfidence. Implementing a time window thus seemed to support children in their evaluation of confidence and led to more accurate monitoring. Our findings indicate that a brief pause where the child can "Stop and Think" can improve not only performance (as was shown in previous studies: [38, 39, 60]), but also monitoring accuracy. Especially children with difficulties inhibiting a prepotent response may benefit from more time [25]. Further, giving time to enhance monitoring accuracy is also in line with recent findings [40] indicating that information processing is not terminated when a decision is made. More information seems to accumulate between a memory decision and the corresponding monitoring judgment, supporting the idea that additional time may lead to more accurate evaluations due to the accumulation of information supervised by monitoring

processes. Even though our experimentally induced manipulation cannot be seen as a congruent and identical method compared to research focusing on a delay, results appear to indicate that the underlying processes are related to each other. In addition to the advantages over "more time to think" known from previous research, the present study discovered an impact on two conceptually different monitoring measures. It is of particular interest that the benefits were not limited to just one aspect of monitoring; instead, our findings might hint at the possibility that monitoring processes overall were affected. This is promising for future research.

Nevertheless, the present study revealed only small effects on the "Stop and Think" manipulation. Perhaps, implementing an extended time window is insufficient to reduce this overoptimistic behavior entirely. Therefore, the possible negative side effects resulting from overconfidence [61], such as ending a learning phase too early or not investing enough time in tasks with increased demands, cannot be completely overcome with such an approach [62, 63]. By giving children more time to consider their answers and their confidence, we could only enable one aspect: providing time for transmission and allowing the individual to be prepared at least on a neurological level. This delay may not be sufficient to fully profit from this neurological readiness; an individual must experience the benefits of monitoring in an environment in which advantages can emerge (for example, asking back when unsure to avoid errors).

Contrary to our assumption, kindergarteners did not disproportionally benefit more from our manipulation. It may be possible that second graders' inhibition skills are at this time not as far developed as monitoring processes may require. Research based on neurological studies indicates that inhibitory control skills' maturation continues until adulthood [64], and the ability to inhibit a prepotent response evolves until adolescence [65]. Therefore, it seems reasonable that second graders equally benefited from our induced support [66]. The spectrum of an individual benefiting from such a "Stop and Think" may be much broader than expected.

When it came to our individual differences approach and our attempt to better understand the role of inhibition for monitoring, our results revealed a significant positive correlation between inhibition and the monitoring discrimination score only for kindergarteners. This relation may indicate that better inhibition skills can indeed be associated with more accurate monitoring skills, especially at an earlier age. However, these findings were not confirmed considering the bias index. Therefore, the relation between inhibition and monitoring accuracy seems still not fully understood. Although other research [21] suggests that accurate monitoring may result from better EF, our results do not reflect the strong interrelation we had expected. The results only displayed a significant correlation for kindergarteners, but the correlations from both age groups were very close in their r values. The insignificant correlation within the second graders may be due to a reduction in the statistical power because of the somewhat smaller sample size in the older age group. Therefore, this insignificant correlation should be interpreted with caution [67]. Results from the present study question strong assumptions that accurate monitoring can be supported through a certain level of inhibition skills. Our findings indicate that inhibition is a necessary but not a sufficient prerequisite for accurate monitoring in children. In her review, Roebers [18] noted that methodical differences are likely to contribute to the typically weak connection between monitoring and inhibitory skills. Capturing in detail and with different methods subcomponents of both constructs may enable to compare different subcomponents. For example, Kälin et al. [21] found a relation between inhibition and implicit, but not explicit measures of monitoring. The association between metacognition and EF could thus vary as a function of inhibition and monitoring measures. In fact, a meta-analysis showed that comparing different tasks for measuring inhibition is specific to a given age range [68]. The utilization of a specific measure must be adapted to a precise age range of interest because, over time, the behavioral manifestation of inhibitory skills changes. This finding highlights the complexity of choosing the right measure for the

correct age range when relating monitoring and inhibition to each other. Recent results suggest that metacognitive skills are far more present in young children (3 to 4 years old) than previously expected and that EF and metacognition are related to each other, underscoring the importance of pursuing research in this direction [26, 43]. Correlational analysis within the experimental group revealed no significant relation between inhibition and monitoring accuracy. These findings suggest that our manipulation does not affect participants differently depending on their inhibitory control abilities.

## Implication

Even though the results yielded only small effects, they shed light on yet not fully understood monitoring processes. Given the theoretical background [18, 69] and the effects of the present study, the evidence supports the idea that additional time for monitoring may indeed result in more accurate monitoring. Also, from a neurological perspective, we would expect that neural signals generated from the ACC transferred to frontal regions need time for transmission [27]. In other words, the metacognitive neurological signal then has time to strengthen and influence monitoring processes. Accurate monitoring is highly relevant for everyday life situations. Not only young children but even adults benefit from more sophisticated monitoring skills. Observing, reviewing, and evaluating the ongoing cognitive processes are essential in the school setting, higher education, and following career [70, 71]. To the best of our knowledge, the present study is the first that tried to increase the time window for children's monitoring, facilitating a transmission for metacognition within an experimental design.

## Limitation

No study is perfect and this one is no exception. Naturally, we cannot be sure that children actively engaged in monitoring during the delay. It is possible that despite our—"Stop and Think"—instruction, the processing of metacognitive signals was not increased. Perhaps for some children, the process of profoundly thinking about their answers can only be achieved if they, for example, have an intrinsic willingness. For future research, an implemented reminder during the animation may trigger cognitive activation for monitoring [72]. Additionally, comparing a delay without any instructions and, therefore, simply allowing more time to reflect in an unguided way would lead to a differentiated understanding of the hidden processes. Based on our and previous findings, the willingness together with additional time to reflect are essential [36, 38].

## Conclusion

The present results indicate that giving young children more time to—"Stop and Think"—can improve monitoring accuracy and reduce overconfidence. Additionally, the outcomes suggest that this time window during which children take time to process and generate their answers and evaluate their confidence in the chosen answer can be strengthened with external support. With a more profound understanding of the underlying processes and how they can be supported, we may help facilitate learning activities for students and support teachers in the school setting.

## Supporting information

**S1 File. Data monitoring.**
(XLSX)

**S2 File. Data inhibition.**
(XLSX)

## Author Contributions

**Conceptualization:** Sophie Wacker, Claudia M. Roebers.

**Formal analysis:** Sophie Wacker.

**Methodology:** Sophie Wacker, Claudia M. Roebers.

**Project administration:** Sophie Wacker, Claudia M. Roebers.

**Supervision:** Claudia M. Roebers.

**Writing – original draft:** Sophie Wacker.

**Writing – review & editing:** Sophie Wacker.

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
