## [Decision Letter · Decision Letter 0]

19 Apr 2022

PONE-D-22-05420Stop and think: Additional time supports monitoring processes in young childrenPLOS ONE

Dear Dr. Wacker,

Thank you for submitting your manuscript to PLOS ONE. After careful consideration, we feel that it has merit but does not fully meet PLOS ONE’s publication criteria as it currently stands. Therefore, we invite you to submit a revised version of the manuscript that addresses the points raised during the review process.

 All three reviewers agree that this research is of high quality, but each make clear and succinct suggestions for how to improve the paper. Comments concern the literature review and conceptual motivation in the introduction, some suggest ways to clarify methods and analysis details, and others point to how to improve interpretation of the analyses. I agree with their suggestions. Please address every comment in the revision.

We look forward to receiving your revised manuscript.

Kind regards,

Micah B. Goldwater, Ph.D

Academic Editor

PLOS ONE

Journal Requirements:

Reviewers' comments:

Reviewer's Responses to Questions

**Comments to the Author**

1. Is the manuscript technically sound, and do the data support the conclusions?

Reviewer #1: Partly

Reviewer #2: Partly

Reviewer #3: Yes

2. Has the statistical analysis been performed appropriately and rigorously? 

Reviewer #1: I Don't Know

Reviewer #2: Yes

Reviewer #3: Yes

3. Have the authors made all data underlying the findings in their manuscript fully available?

Reviewer #1: Yes

Reviewer #2: No

Reviewer #3: No

4. Is the manuscript presented in an intelligible fashion and written in standard English?

Reviewer #1: Yes

Reviewer #2: Yes

Reviewer #3: Yes

5. Review Comments to the Author

Reviewer #1: In this manuscript Wacker and Roebers investigated the discrepancy between children’s monitoring abilities in day-to-day life, which appear to be pretty good, and their monitoring abilities in research settings, which appear to be poor. They tested if giving children extra time before self-reporting their monitoring (i.e., confidence judgements) increased monitoring accuracy in a research setting and if these self-reports were related to inhibitory control. The manuscript was succinct and straightforward and made it clear that this study was motivated by previous research. Furthermore, the ideas/predictions were easy to follow. I only have a few clarifications or methodological/statistical related comments.

1. One major issue is that the data appear to be all between subjects (based on the methods section) and yet some of the information in the results section makes it seem like some data was being treated as within subjects. I’m assuming this is a typo and that the correct test was run (degrees of freedom numbers in results section suggest a between subjects ANOVA), but it does need to be corrected in the text.

• Methods section text: “Children were randomly assigned to either the control group (CG); they solved the task as described above or the experimental group (EG).”

• Fig 1 and Fig 2 text: “Note. Discrimination Score, between factors Age (Kindergartners vs. Second Graders), within factors Condition (Control Group (CG) vs. Experimental Group (EG))”

Related to this issue, please indicate what type of ANOVA you are running (between-subjects, within-subjects, or mixed) when you describe your ANOVA based analyses (currently the manuscript only says that it was a 2x2 design).

2. The last sentence in the abstract, “The pattern of results revealed an interactive role between monitoring and inhibition only for children in kindergarten and not for second graders”, is not fully supported by the results. There was a correlation done between monitoring and inhibitory control for each age group and so you could say that there was a correlation between monitoring and inhibition for kindergarteners. However, unless you directly test an interaction between monitoring and inhibitory control and include both age groups in this test, you cannot say that there was an interaction between monitoring and inhibitory control only for one age group.

• Nieuwenhuis, S., Forstmann, B. & Wagenmakers, EJ. Erroneous analyses of interactions in neuroscience: a problem of significance. Nat Neurosci 14, 1105–1107 (2011). https://doi.org/10.1038/nn.2886

Moreover, the correlations between monitoring and inhibitory control were very close in their r values (0.225 for kindergarteners and 0.200 for second graders) despite them only being significant for kindergarteners (p<.05 vs p=.104). An article by Tamar and Jean-Jacques (2019) suggests that a lack of significance can sometimes be due to the sample being underpowered and researchers should be cautious in over interpreting insignificant results. In their article, they do say that a small effect size means an effect is unlikely to be theoretically meaningful. However, I bring up their article because the r values are so close in size between these two groups and I noticed that the kindergarteners are a larger sample than the second graders (n=88 and n=67 respectively). It makes me wonder if the effect for the second graders was non-significant only because the group was smaller and therefore the statistical power was reduced compared to the kindergarten group.

• Makin, T. R., & de Xivry, J. J. O. (2019). Science Forum: Ten common statistical mistakes to watch out for when writing or reviewing a manuscript. Elife, 8, e48175. DOI: 10.7554/eLife.48175

3. I also have a minor suggestion concerning how the ACC is discussed in the introduction.

“To engage in monitoring, the responsible neural networks need time to loop signals from the anterior cingulate cortex (ACC) to frontal structures [25] and to trigger a feeling of uncertainty which may lead the child to overthink his confidence judgments.”

When talking about neural networks and signals, it would be better to keep the description simple and not make conjecture about what those neural signals mean in terms of what the subject is thinking or feeling.

4. The text has minor grammatical errors throughout. Nothing major, but I recommend proofreading again before re-submission.

Reviewer #2: Overall, I think the study provides some interesting data concerning the effect of delaying CR on monitoring accuracy. That being said, this is a pretty well-worn effect in the non-developmental metacognition literature and that should be better acknowledged in the paper.

Introduction

There’s a pretty large literature on the accuracy of delayed metacognitive ratings compared to immediate – see the delayed judgment of learning effect (Nelson & Dunlosky, 1991) - which hasn’t been addressed in the introduction and (at least from a non-developmental perspective) might limit the novelty of the findings

The other literature that seems relevant but isn’t included is the idea of confidence ratings being informed by post-decision accumulation of evidence (e.g., Navajas et al., 2016)– so rather than improved accuracy as a result of inhibiting the CR by a short time, the accuracy might improve because the evidence in favour or against the decision has had longer to accumulate

Results

Could the authors include the performance results as a function of experimental group, it would be good to rule out whether the ‘stop-and-think’ procedure improved performance (which might inflate the monitoring accuracy if kids tend to be overconfident). It seems to indicate in the discussion that stop-and-think can impact performance so better monitoring accuracy due to an artefact of better performance needs to be ruled out

Unless I misunderstood both groups completed the inhibition task, so why not include both groups and look at the interaction between group X inhibition predicting accuracy. It would, for example, tell you whether the intervention was more effective for children with low inhibition

Discussion

A few of the points above should also be incorporated into the discussion, especially when trying to outline possible mechanisms why more time is helpful for metacognition

In terms of the limitation you raise, as a future study, it might be interesting to think about the stop-and-think vs some other delay – so rather than explicitly cueing more metacognitive thought simply allowing more time to reflect in an unguided way

Minor

Line 75 – I would remove the tail end of the sentence “according to the often used instruction ‘stop and think’”, it’s not necessarily a familiar manipulation to people not in this sub-field

Report the exact p-value on line 233, line 248, line 264 etc (i.e. don’t write p < .05)

For the figures, need a description of the what the boxes indicate. I would add a dotted horizontal line at the zero mark

The figure captions indicate that the error bars are standard errors of the mean. Is this correct? – they seem extremely wide from my cursory glance if the SDs reported in paper are correct, the SE should be smaller than the SD, if I remember correctly)

Reviewer #3: This paper investigates the role of inhibition on metacognitive monitoring, taking two approaches. The authors manipulated whether participants received a delay before making a confidence judgment, finding that children showed greater monitoring accuracy after the “Stop and Think” manipulation. In addition, the authors found that individual differences in inhibitory control were related to metacognitive monitoring accuracy for younger children but not older children. I believe this study addresses a theoretically important issue - understanding the relation between metacognitive monitoring and executive function, as well as the development of this relation.

Overall, the paper is clear and well-written. Below I outline some questions and suggestions to enhance the clarity of the paper:

The authors use binary pronouns throughout the paper, with multiple references to “him/his” without any other gender option. Use of binary pronouns can exclude those who are non-binary or who hold other gender identities. I would encourage the authors to use gender neutral language (they/them/theirs) throughout the paper. The APA supports the use of the singular they in academic writing.

In the method section, the authors mention that the Hearts and Flowers task was used to measure inhibition and cognitive flexibility. However, no description of this task is offered. I would encourage the authors to include a detailed description of the task protocol.

In the results section (line 206-207), the authors mention that participants were excluded if overall accuracy was lower than .50. This is presumably because that would show a response bias in the task. I would encourage the authors to spell that out for the reader.

On a theoretical level, it is possible that imposing a greater delay encourages further processing in the base-level task, rather than greater deployment of metacognitive processes. Further processing in the base-level task may make metacognition easier, perhaps reducing the effort required to monitor the base-level task. In other words, the delay may contribute to a stronger signal from the base-level task, which would require less metacognitive effort to detect. I would encourage the authors to consider this possibility in their revisions.

There were a couple of sentences where the writing/word choice was unclear:

On lines 90-92, the sentence ends with, “... which may lead the child to overthink his confidence judgments.” The use of “overthink” often has negative connotations, but I think the authors are trying to describe a situation in which typical monitoring is occurring.

On lines 341-342, the authors write, “A meta-analysis showed that comparing different tasks for measuring inhibitory control is specific to a given age range.” This is unclear and could be easily clarified by including more detail about this finding.

6. PLOS authors have the option to publish the peer review history of their article (what does this mean?). If published, this will include your full peer review and any attached files.

Reviewer #1: No

Reviewer #2: No

Reviewer #3: No

---

## [Author Response · Author response to Decision Letter 0]

3 Jun 2022

Response to Reviewer #1

My comments appear in blue

Reviewer #1: In this manuscript Wacker and Roebers investigated the discrepancy between children’s monitoring abilities in day-to-day life, which appear to be pretty good, and their monitoring abilities in research settings, which appear to be poor. They tested if giving children extra time before self-reporting their monitoring (i.e., confidence judgements) increased monitoring accuracy in a research setting and if these self-reports were related to inhibitory control. The manuscript was succinct and straightforward and made it clear that this study was motivated by previous research. Furthermore, the ideas/predictions were easy to follow. I only have a few clarifications or methodological/statistical related comments.

Thank you for your helpful comments and for giving me the opportunity to revise the manuscript. I have been able to understand your inputs and adapted them accordingly. You’ll find my answers to your comments in blue.

1. One major issue is that the data appear to be all between subjects (based on the methods section) and yet some of the information in the results section makes it seem like some data was being treated as within subjects. I’m assuming this is a typo and that the correct test was run (degrees of freedom numbers in results section suggest a between subjects ANOVA), but it does need to be corrected in the text.

• Methods section text: “Children were randomly assigned to either the control group (CG); they solved the task as described above or the experimental group (EG).”

• Fig 1 and Fig 2 text: “Note. Discrimination Score, between factors Age (Kindergartners vs. Second Graders), within factors Condition (Control Group (CG) vs. Experimental Group (EG))”

Related to this issue, please indicate what type of ANOVA you are running (between-subjects, within-subjects, or mixed) when you describe your ANOVA based analyses (currently the manuscript only says that it was a 2x2 design).

Thank you for your comments. I have added some more details about the design and the type of the measures (ANOVA) and adjusted the descriptions of the figures (statistical analysis and result section).

2. The last sentence in the abstract, “The pattern of results revealed an interactive role between monitoring and inhibition only for children in kindergarten and not for second graders”, is not fully supported by the results. There was a correlation done between monitoring and inhibitory control for each age group and so you could say that there was a correlation between monitoring and inhibition for kindergarteners. However, unless you directly test an interaction between monitoring and inhibitory control and include both age groups in this test, you cannot say that there was an interaction between monitoring and inhibitory control only for one age group.

• Nieuwenhuis, S., Forstmann, B. & Wagenmakers, EJ. Erroneous analyses of interactions in neuroscience: a problem of significance. Nat Neurosci 14, 1105–1107 (2011). https://doi.org/10.1038/nn.2886

Moreover, the correlations between monitoring and inhibitory control were very close in their r values (0.225 for kindergarteners and 0.200 for second graders) despite them only being significant for kindergarteners (p<.05 vs p=.104). An article by Tamar and Jean-Jacques (2019) suggests that a lack of significance can sometimes be due to the sample being underpowered and researchers should be cautious in over interpreting insignificant results. In their article, they do say that a small effect size means an effect is unlikely to be theoretically meaningful. However, I bring up their article because the r values are so close in size between these two groups and I noticed that the kindergarteners are a larger sample than the second graders (n=88 and n=67 respectively). It makes me wonder if the effect for the second graders was non-significant only because the group was smaller and therefore the statistical power was reduced compared to the kindergarten group.

• Makin, T. R., & de Xivry, J. J. O. (2019). Science Forum: Ten common statistical mistakes to watch out for when writing or reviewing a manuscript. Elife, 8, e48175. DOI: 10.7554/eLife.48175

Because our data collection was carried out during Covid-19, due to children in quarantine and the corresponding dropout, we could not capture the H&F task from everyone. Therefore, these considerations addressing unequal sample size and the interpretation of insignificant results are interesting, and I have included these reflections in the discussion (lines: 392 - 396). Thank you for this input. I have also adjusted the abstract section (Lines: 34-36).

3. I also have a minor suggestion concerning how the ACC is discussed in the introduction.

“To engage in monitoring, the responsible neural networks need time to loop signals from the anterior cingulate cortex (ACC) to frontal structures [25] and to trigger a feeling of uncertainty which may lead the child to overthink his confidence judgments.”

When talking about neural networks and signals, it would be better to keep the description simple and not make conjecture about what those neural signals mean in terms of what the subject is thinking or feeling.

Good point, thank you. Mainly because we could not measure cognitive signals during our task, we must be careful in interpreting such potential neurological signals and their relation to other cognitive concepts. 

Nevertheless, ample evidence shows that frontal structures and the ACC may be seen as a neurological correlate of monitoring. We also never tried to describe this relation and neurological transmission as a fixed process for the feeling of uncertainty; instead, we used the words like "may" to underline this possible neurological transmission under consideration of the neurological studies. However, I agree that the application and the description should be discussed more carefully, and this association may be expressed simpler. I have adjusted the corresponding part (Lines: 91-96).

4. The text has minor grammatical errors throughout. Nothing major, but I recommend proofreading again before re-submission.

Thank you for your advice. The re-submission has now been proofread again.

Response to Reviewer #2

My comments appear in blue

Reviewer #2: Overall, I think the study provides some interesting data concerning the effect of delaying CR on monitoring accuracy. That being said, this is a pretty well-worn effect in the non-developmental metacognition literature and that should be better acknowledged in the paper.

Thank you for your comments and the possibility to review this paper.

Introduction

There’s a pretty large literature on the accuracy of delayed metacognitive ratings compared to immediate – see the delayed judgment of learning effect (Nelson & Dunlosky, 1991) - which hasn’t been addressed in the introduction and (at least from a non-developmental perspective) might limit the novelty of the findings

Even though Nelsons and Dunlosky's study differs from our design, their results support indirect evidence that time may play a crucial role in monitoring accuracy. I have included their results in the introduction section. Thank you for this comment. (Lines: 103 - 106).

The other literature that seems relevant but isn’t included is the idea of confidence ratings being informed by post-decision accumulation of evidence (e.g., Navajas et al., 2016)– so rather than improved accuracy as a result of inhibiting the CR by a short time, the accuracy might improve because the evidence in favour or against the decision has had longer to accumulate

The study from Navajas et al. (2019) revealed exciting insights between giving an answer and rating a confidence judgment on this answer. I would argue that it may also be an interactive role - while inhibition is needed to take more time to allow further processing and accumulation, monitoring processes may supervise these procedures. Because monitoring processes have to be active to detect and make use of the added information during this interval because without - there may be a change, this potential of further processing may not even be detected. I have integrated their findings in the introduction and the discussion section. (Lines: 114 – 121, 352 – 359)

Results

Could the authors include the performance results as a function of experimental group, it would be good to rule out whether the ‘stop-and-think’ procedure improved performance (which might inflate the monitoring accuracy if kids tend to be overconfident). It seems to indicate in the discussion that stop-and-think can impact performance so better monitoring accuracy due to an artefact of better performance needs to be ruled out

Thank you for this comment. In the result section, I added the corresponding analysis addressing performance accuracy as a post hoc test. (Lines: 252 – 259).

Unless I misunderstood both groups completed the inhibition task, so why not include both groups and look at the interaction between group X inhibition predicting accuracy. It would, for example, tell you whether the intervention was more effective for children with low inhibition

Both the EG and the CG have completed the H&F task. In our comparison, however, we intentionally compared only the inhibition of the CG with the MC task because we were interested in exploring whether individual differences in inhibition are related to monitoring accuracy independently from our manipulation. However, I agree that the inclusion of the EG may reveal interesting additional information. This allows us to address whether inserting the delay between recognition and monitoring has a differential effect on participants depending on their inhibitory control skills. I included the analysis in the results section (Lines: 259 – 269, 319 – 325).

Discussion

A few of the points above should also be incorporated into the discussion, especially when trying to outline possible mechanisms why more time is helpful for metacognition

Information added.

In terms of the limitation you raise, as a future study, it might be interesting to think about the stop-and-think vs some other delay – so rather than explicitly cueing more metacognitive thought simply allowing more time to reflect in an unguided way

Information added. (Lines: 434 – 436)

Minor

Line 75 – I would remove the tail end of the sentence “according to the often used instruction ‘stop and think’”, it’s not necessarily a familiar manipulation to people not in this sub-field

I have adjusted the corresponding part.

Report the exact p-value on line 233, line 248, line 264 etc (i.e. don’t write p < .05)

The exact p values have been added.

For the figures, need a description of the what the boxes indicate. I would add a dotted horizontal line at the zero mark

I have added the information.

The figure captions indicate that the error bars are standard errors of the mean. Is this correct? – they seem extremely wide from my cursory glance if the SDs reported in paper are correct, the SE should be smaller than the SD, if I remember correctly).

Thank you for bringing this to my attention. I have adjusted the description of the legend. The visualization displays a boxplot with corresponding whiskers. The whiskers indicate the min/max value within the definition of a whisker's length (1.5* interquartile range) for the dependent variables. Data points outside of this interquartile range are defined as outliers.

Response to Reviewer #3

My comments appear in blue

Reviewer #3: This paper investigates the role of inhibition on metacognitive monitoring, taking two approaches. The authors manipulated whether participants received a delay before making a confidence judgment, finding that children showed greater monitoring accuracy after the “Stop and Think” manipulation. In addition, the authors found that individual differences in inhibitory control were related to metacognitive monitoring accuracy for younger children but not older children. I believe this study addresses a theoretically important issue - understanding the relation between metacognitive monitoring and executive function, as well as the development of this relation.

Thank you for your positive feedback addressing the importance of our study.

Overall, the paper is clear and well-written. Below I outline some questions and suggestions to enhance the clarity of the paper:

The authors use binary pronouns throughout the paper, with multiple references to “him/his” without any other gender option. Use of binary pronouns can exclude those who are non-binary or who hold other gender identities. I would encourage the authors to use gender neutral language (they/them/theirs) throughout the paper. The APA supports the use of the singular they in academic writing.

Thank you for bringing this to my attention. I have changed the pronouns to a gender-neutral language.

In the method section, the authors mention that the Hearts and Flowers task was used to measure inhibition and cognitive flexibility. However, no description of this task is offered. I would encourage the authors to include a detailed description of the task protocol.

Because our primary interest lies in the measurement of monitoring, and we created a new task capturing monitoring skills, we wanted to ensure that the procedure of this novel task was described in detail. The Hearts and Flowers Task was already applied several times in other studies. Because of the frequent use in the research field of executive functions, we diminished the Hearts and Flowers task's description in our method section. For replication, we indicated the primary literature to provide the connection to the studies using the Hearts and Flowers task. However, I agree that it may confuse the reader why one task is deeply explained and the other not. I, therefore, inserted a short description of the Hearts and Flowers task, especially with additional information about technical and methodological details. (Lines: 229 – 235).

In the results section (line 206-207), the authors mention that participants were excluded if overall accuracy was lower than .50. This is presumably because that would show a response bias in the task. I would encourage the authors to spell that out for the reader.

Information added.

On a theoretical level, it is possible that imposing a greater delay encourages further processing in the base-level task, rather than greater deployment of metacognitive processes. Further processing in the base-level task may make metacognition easier, perhaps reducing the effort required to monitor the base-level task. In other words, the delay may contribute to a stronger signal from the base-level task, which would require less metacognitive effort to detect. I would encourage the authors to consider this possibility in their revisions.

Thank you for this interesting comment. I have integrated your input in the introduction and discussion section. (Lines: 114-121, 352-359).

There were a couple of sentences where the writing/word choice was unclear:

On lines 90-92, the sentence ends with, “... which may lead the child to overthink his confidence judgments.” The use of “overthink” often has negative connotations, but I think the authors are trying to describe a situation in which typical monitoring is occurring.

Good point! We do not associate "overthinking" with a negative context. We wanted to describe how this "stop and think" condition may trigger a feeling of uncertainty where children may hesitate and "overthink" or "reassess" their given confidence judgment. The term "overthink" describes monitoring processes where children evaluate their answers and give them a second thought.

On lines 341-342, the authors write, “A meta-analysis showed that comparing different tasks for measuring inhibitory control is specific to a given age range.” This is unclear and could be easily clarified by including more detail about this finding.

I added some information for clarification.

---

## [Decision Letter · Decision Letter 1]

15 Jul 2022

PONE-D-22-05420R1Stop and think: Additional time supports monitoring processes in young childrenPLOS ONE

Dear Dr. Wacker,

Thank you for submitting your manuscript to PLOS ONE. All the reviewers and I agree you have improved the manuscript from the first version. Two of three think the manuscript is ready to be accepted, but one reviewer has outlined how to further clarify your arguments. Therefore, we invite you to submit a revised version of the manuscript that addresses these points (see below). 

We look forward to receiving your revised manuscript.

Kind regards,

Micah B. Goldwater, Ph.D

Academic Editor

PLOS ONE

Journal Requirements:

Reviewers' comments:

Reviewer's Responses to Questions

**Comments to the Author**

1. If the authors have adequately addressed your comments raised in a previous round of review and you feel that this manuscript is now acceptable for publication, you may indicate that here to bypass the “Comments to the Author” section, enter your conflict of interest statement in the “Confidential to Editor” section, and submit your "Accept" recommendation.

Reviewer #1: All comments have been addressed

Reviewer #2: All comments have been addressed

Reviewer #3: (No Response)

2. Is the manuscript technically sound, and do the data support the conclusions?

Reviewer #1: Yes

Reviewer #2: Yes

Reviewer #3: Yes

3. Has the statistical analysis been performed appropriately and rigorously? 

Reviewer #1: Yes

Reviewer #2: Yes

Reviewer #3: Yes

4. Have the authors made all data underlying the findings in their manuscript fully available?

Reviewer #1: Yes

Reviewer #2: Yes

Reviewer #3: Yes

5. Is the manuscript presented in an intelligible fashion and written in standard English?

Reviewer #1: Yes

Reviewer #2: Yes

Reviewer #3: Yes

6. Review Comments to the Author

Reviewer #1: All of my previous comments/suggestions have been addressed. Thank you to the authors for their edits and replies to my comments/suggestions. I believe the manuscript should be accepted for publication.

Reviewer #2: The authors have addressed all of the comments from my earlier review. The paper should make an interesting contribution to the literature

Reviewer #3: I think that the authors responded to all of the reviewers' comments from the original submission. The research is sound, but the manuscript still has some issues in terms of clarity and organization. Below, I outline some suggestions for improvement:

In lines 52-56, the authors introduce the process of monitoring, and equate it with confidence and performance estimation. The authors should be clear that these are just 2 examples, and that there are other forms of monitoring (e.g., judgments of learning, feeling of knowing, difficulty estimations, etc.).

In the paragraph that spans lines 91-121, the argument the authors are trying to make is unclear. They discuss neurological underpinnings of monitoring, indirect evidence of this, the role of increased time on performance and accuracy, and metacognitive processing post-decision. It seems that the paragraph should be organized to make a more direct argument, or it may be split into several paragraphs to make several different arguments

In lines 99-101, the authors suggest that part of the monitoring process involves asking "Am I really sure about my answer?" Previous work has investigated both explicit and implicit forms of metacognitive monitoring, so the authors should clarify their viewpoint on the role of conscious processing in metacognitive monitoring in young children

In addition, in the discussion of previous research on what happens during the "time to loop signals from the ACC to frontal structures", I suggest the authors clarify what is happening functionally during this process, and what the average processing time is, as these seem relevant to the manipulation in the current study

The authors write, "The engagement with uncertainty may trigger cognitive processing." This is unclear, did the authors potentially mean "may trigger metacognitive processing"?

Later in the paper, the authors write, "Inhibiting the prepotent response allowing neurological signals to spread out…" What is meant by "spread out"? I noticed that there were no citations next to this claim - what is the previous research that suggests this?

In line 124, the authors write, "stop and think! About the answer profoundly". As they mention later in the manuscript, it is not guaranteed that participants thought about the answer profoundly, so I would encourage the authors to reconsider the wording here.

In line 127, the authors should clarify whether they believe the manipulation will positively affect children's monitoring skills (implying the acquisition of new strategies and potentially stable improvement) or their monitoring performance (implying temporary, in the moment improvements).

In line 134, the authors write, "We did not expect any effect on recognition". At this point in the manuscript the recognition task has not been described. I would suggest a slightly more detailed description of the base-level task in this paragraph.

In lines 177-178, the authors write "Additionally, during one test session, N = 20 children had to quarantine due to COVID-19." I would encourage the authors to explain why these children needed to be excluded. Was it due to incomplete data/ Which portions of the experiment did they complete or fail to complete?

One interesting finding was that there was a correlation between monitoring and inhibition in kindergarteners only in the control condition. I suggest that the authors provide a potential explanation for this finding in the Discussion section.

In lines 336-338, the authors write, "In absolute terms, children were able to discriminate substantially between correct and incorrect responses". Was there an analysis described in the Results section that directly tested this? Were the discrimination scores significantly different than 0?

In the Implications paragraph, I think the authors could include a more detailed and compelling description of the real-world implications of "stopping to think." Where might this come into play for both children and adults? In addition, the sentences "Also, from a neurological perspective, we would expect that neural signals generated from the ACC transferred to frontal regions need time for transmission. In other words, the metacognitive neurological signal then has time to spread out and influence monitoring processes." do not seem to be an implication of the current study, but speculation about the neurological processes involved. While appropriate for the discussion section, I do not think this belongs in the implications section.

7. PLOS authors have the option to publish the peer review history of their article (what does this mean?). If published, this will include your full peer review and any attached files.

Reviewer #1: No

Reviewer #2: **Yes: **Kit Spenser Double

Reviewer #3: No

---

## [Author Response · Author response to Decision Letter 1]

10 Aug 2022

Thank you for the input addressing my paper in order to clarify and to improve the quality of the present study.

My Comments appear in blue.

Response to Reviewer #1 and #2

Reviewer #1: All of my previous comments/suggestions have been addressed. Thank you to the authors for their edits and replies to my comments/suggestions. I believe the manuscript should be accepted for publication.

Reviewer #2: The authors have addressed all of the comments from my earlier review. The paper should make an interesting contribution to the literature.

Thank you for your positive feedback!

Response to Reviewer #3

Reviewer #3: I think that the authors responded to all of the reviewers' comments from the original submission. The research is sound, but the manuscript still has some issues in terms of clarity and organization. Below, I outline some suggestions for improvement:

Thank you for the acknowledgment of my improvements. I thank you for your further recommendations to improve the quality of my paper.

In lines 52-56, the authors introduce the process of monitoring, and equate it with confidence and performance estimation. The authors should be clear that these are just 2 examples, and that there are other forms of monitoring (e.g., judgments of learning, feeling of knowing, difficulty estimations, etc.).

I have included a paragraph to draw the reader's attention to the fact that there are also some other measures of monitoring processes (lines: 53-57).

In the paragraph that spans lines 91-121, the argument the authors are trying to make is unclear. They discuss neurological underpinnings of monitoring, indirect evidence of this, the role of increased time on performance and accuracy, and metacognitive processing post-decision. It seems that the paragraph should be organized to make a more direct argument, or it may be split into several paragraphs to make several different arguments

Thank you for your comment. I organized and split the part into several paragraphs to enhance clearness (lines: 95 – 128).

In lines 99-101, the authors suggest that part of the monitoring process involves asking "Am I really sure about my answer?" Previous work has investigated both explicit and implicit forms of metacognitive monitoring, so the authors should clarify their viewpoint on the role of conscious processing in metacognitive monitoring in young children

I have added some more information (line: 106 - 109).

In addition, in the discussion of previous research on what happens during the "time to loop signals from the ACC to frontal structures", I suggest the authors clarify what is happening functionally during this process, and what the average processing time is, as these seem relevant to the manipulation in the current study

The procedure and measure section explains how long we set our delay (lines: 217 - 223). Because this is a novel approach, and to our knowledge, no studies so fare exist addressing this issue, we do not have any additional references. Our delay builds on the idea of neurological diffusion (about 200-250 ms) as well on studies implementing a delay in other cognitive tasks. We argue that firstly additional time for diffusion of neurological signals and secondly time for monitoring processes result in a reasonable delay. Further research is needed to identify the best processing time range. Nevertheless, our study contributes the first insight regarding a delay for monitoring and may be a guiding reference point for further studies.

The authors write, "The engagement with uncertainty may trigger cognitive processing." This is unclear, did the authors potentially mean "may trigger metacognitive processing"?

I have added some information for clarification (lines: 108 – 109).

Later in the paper, the authors write, "Inhibiting the prepotent response allowing neurological signals to spread out…" What is meant by "spread out"? I noticed that there were no citations next to this claim - what is the previous research that suggests this?

By using the term "spread out" we are addressing the diffusion of neurological signals. These neurological signals emerge at a point x and subsequently diffuse or, as we call it, "spread out".

I have changed the wording and I also added citations (lines: 126 – 128).

In line 124, the authors write, "stop and think! About the answer profoundly". As they mention later in the manuscript, it is not guaranteed that participants thought about the answer profoundly, so I would encourage the authors to reconsider the wording here.

Thank you for your comment. I have adapted the corresponding part (line: 131).

In line 127, the authors should clarify whether they believe the manipulation will positively affect children's monitoring skills (implying the acquisition of new strategies and potentially stable improvement) or their monitoring performance (implying temporary, in the moment improvements).

I added that we are only expecting temporary and, therefore, in the moment improvements (line: 138). However, further research is needed to evaluate if more practice with a "Stop and Think" condition would also lead to an acquisition of a “Stop and Think” as a metacognitive strategy and be also transferable to other domains.

In line 134, the authors write, "We did not expect any effect on recognition". At this point in the manuscript the recognition task has not been described. I would suggest a slightly more detailed description of the base-level task in this paragraph.

I have added some information regarding the task (lines: 131 –136).

In lines 177-178, the authors write "Additionally, during one test session, N = 20 children had to quarantine due to COVID-19." I would encourage the authors to explain why these children needed to be excluded. Was it due to incomplete data/ Which portions of the experiment did they complete or fail to complete?

These children did not solve the paired associate learning task. Due to the restrictions in the school setting we were not allowed to retest them. 

I have added some information (lines: 189 – 191).

One interesting finding was that there was a correlation between monitoring and inhibition in kindergarteners only in the control condition. I suggest that the authors provide a potential explanation for this finding in the Discussion section.

These results are discussed in the discussion section (lines: 398 - 430).

In lines 336-338, the authors write, "In absolute terms, children were able to discriminate substantially between correct and incorrect responses". Was there an analysis described in the Results section that directly tested this? Were the discrimination scores significantly different than 0?

Yes, this marked sentence is one of our main findings regarding monitoring. The corresponding analysis is in the result section (ANOVA with dependent variable: discrimination score). Discrimination of confidence judgments between correctly and incorrectly solved items) and the bias index (under- and overestimation).

In the Implications paragraph, I think the authors could include a more detailed and compelling description of the real-world implications of "stopping to think." Where might this come into play for both children and adults? In addition, the sentences "Also, from a neurological perspective, we would expect that neural signals generated from the ACC transferred to frontal regions need time for transmission. In other words, the metacognitive neurological signal then has time to spread out and influence monitoring processes." do not seem to be an implication of the current study, but speculation about the neurological processes involved. While appropriate for the discussion section, I do not think this belongs in the implications section.

I have added some information (lines: 438 - 441).

---

## [Decision Letter · Decision Letter 2]

30 Aug 2022

Stop and think: Additional time supports monitoring processes in young children

PONE-D-22-05420R2

Dear Dr. Wacker,

We’re pleased to inform you that your manuscript has been judged scientifically suitable for publication and will be formally accepted for publication once it meets all outstanding technical requirements.

Kind regards,

Micah B. Goldwater, Ph.D

Academic Editor

PLOS ONE

Additional Editor Comments (optional):

Reviewers' comments:

Reviewer's Responses to Questions

**Comments to the Author**

1. If the authors have adequately addressed your comments raised in a previous round of review and you feel that this manuscript is now acceptable for publication, you may indicate that here to bypass the “Comments to the Author” section, enter your conflict of interest statement in the “Confidential to Editor” section, and submit your "Accept" recommendation.

Reviewer #3: All comments have been addressed

2. Is the manuscript technically sound, and do the data support the conclusions?

Reviewer #3: Yes

3. Has the statistical analysis been performed appropriately and rigorously? 

Reviewer #3: Yes

4. Have the authors made all data underlying the findings in their manuscript fully available?

Reviewer #3: Yes

5. Is the manuscript presented in an intelligible fashion and written in standard English?

Reviewer #3: Yes

6. Review Comments to the Author

Reviewer #3: A big thank you to the authors for addressing my comments. I believe the manuscript should be accepted for publication.

7. PLOS authors have the option to publish the peer review history of their article (what does this mean?). If published, this will include your full peer review and any attached files.

Reviewer #3: No

---

## [Editor Report · Acceptance letter]

5 Sep 2022

PONE-D-22-05420R2 

Stop and think: Additional time supports monitoring processes in young children 

Dear Dr. Wacker:

I'm pleased to inform you that your manuscript has been deemed suitable for publication in PLOS ONE. Congratulations! Your manuscript is now with our production department. 

Kind regards, 

on behalf of

Dr. Micah B. Goldwater 

Academic Editor

PLOS ONE